# Investigation of Lipolytic-Secreting Bacteria from an Artificially Polluted Soil Using a Modified Culture Method and Optimization of Their Lipase Production

**DOI:** 10.3390/microorganisms9122590

**Published:** 2021-12-15

**Authors:** Van Hong Thi Pham, Jaisoo Kim, Soonwoong Chang, Woojin Chung

**Affiliations:** 1Department of Environmental Energy Engineering, Graduate School of Kyonggi University, Suwon 16227, Korea; vanhtpham@gmail.com; 2Department of Life Science, College of Natural Science of Kyonggi University, Suwon 16227, Korea; jkimtamu@kyonggi.ac.kr; 3Department of Environmental Energy Engineering, College of Creative Engineering of Kyonggi University, Suwon 16227, Korea

**Keywords:** lipase-producing bacteria, modified dependent-culture method, lipolytic bacteria

## Abstract

Compared to lipases from plants or animals, microbial lipases play a vital role in different industrial applications and biotechnological perspectives due to their high stability and cost-effectiveness. Therefore, numerous lipase producers have been investigated in a variety of environments in the presence of lipidic carbon and organic nitrogen sources. As a step in the development of cultivating the unculturable functional bacteria in this study, the forest soil collected from the surrounding plant roots was used to create an artificially contaminated environment for lipase-producing bacterial isolation. The ten strongest active bacterial strains were tested in an enzyme assay supplemented with metal ions such as Ca^2+^, Zn^2+^, Cu^2+^, Fe^2+^, Mg^2+^, K^+^, Co^2+^, Mn^2+^, and Sn^2+^ to determine bacterial tolerance and the effect of these metal ions on enzyme activity. Lipolytic bacteria in this study tended to grow and achieved a high lipase activity at temperatures of 35–40 °C and at pH 6–7, reaching a peak of 480 U/mL and 420 U/mL produced by *Lysinibacillus* PL33 and *Lysinibacillus* PL35, respectively. These potential lipase-producing bacteria are excellent candidates for large-scale applications in the future.

## 1. Introduction

Lipases, which belong to the family of triacylglycerol ester hydrolase, can catalyze the hydrolysis and synthesis of esters formed from glycerol and long-chain fatty acids without the addition of cofactors [1]. Due to their stability in organic solvents, lipases are listed as the third largest group of commercialized enzymes after protease and carbohydrase [2,3]. Recognized for their significance, lipases have unique characteristics that assist in interacting at the interface between aqueous and non-aqueous media. It has been demonstrated that it can be utilized in a wide spectrum of substrates and is highly stable under extreme conditions of temperature, pH, and organic solvents [4].

Compared to lipases from plants and animals, microbial lipases are increasingly valuable and preferable due to the diverse substrates used, high yield production, greater stability, shorter generation time, simple genetic manipulations, and lower production costs. Moreover, microbial lipases as multipurpose biological enzymes are of significant interest for various applications in biological and industrial processes including food and drink, leather, detergents, cosmetic, textile, agrochemicals, the pharmaceutical industries, and waste treatment [5,6,7,8,9]. For instance, lipolytic enzymes secreted by members of the *Bacillus* genus are of interest in biotechnology [10,11]. The presence of oil waste was described as the cause of death of aquatic organisms since oil inhibits the diffusion of oxygen into water. Therefore, bacterial lipase plays a significant role in reducing the oil and fat components in wastewater [12,13].

Numerous bacteria have been investigated as lipase producers, which were isolated from diverse habitats such as oil-contaminated soil, industrial waste, dairy plants, oil-processing factories [14], and from compost heaps [15]. Among these, the commercially valuable ones belong to *Achromobacter, Alcaligenes, Arthrobacter, Bacillus, Burkholderia*, and *Pseudomonas* [16]. However, bacteria that produce multi-enzymes, including lipase, have been explored in other diverse environments. In particular, soil is still an ideal source for unknown functional microorganisms; the unexplored 99% of microbial communities are yet to be cultured in the laboratory [17]. In the previous study, the lipase activity of microorganisms was also investigated in soil observed the higher activity compared to it in water [18].

In the present study, lipolytic bacteria were identified using a modified dependent-culture method to attract multifunctional bacteria from soil. The growth and lipase production of the strong multi-enzyme-producing bacterial strains were optimized prior to its evaluation at extreme conditions. The most promising candidates will be applied to the treatment process of food waste leachate.

## 2. Materials and Methods

### 2.1. Isolation of Lipase-Producing Bacteria

Soil samples were collected from the root-surrounding area of the forest soil at Kyonggi University, Suwon, South Korea. One of the targets in the isolation steps was to encourage bacteria that are capable of tolerating extreme and contaminated environmental conditions such as heavy metal-contaminated, oil-contaminated, and plastic-contaminated soil and water. Therefore, a modified cultivation method was artificially designed to mimic polluted environmental conditions. Ten grams of soil was added to 100 mL of artificial medium containing (g/L): NaCl 10; Cd^2+^ 0.005; Cu^2+^ 0.01, Fe^2+^ 0.01, Mn^2+^ 0.01, zipper bag 0.5 (Low Density Polyethylene- LDPE), and olive oil (10 mL), and was stirred at 120 rpm and ambient temperature for one month. One milliliter of the surviving bacteria in the culture was transferred to fresh LB medium containing autoclaved extracted soil (AES) prepared by adding 100 g soil to 1000 mL distilled water, which was adjusted to a pH of 7 before autoclaving [19]. Pure colonies were isolated and streaked on agar plates for use in further steps.

### 2.2. Screening of Microbial Lipase Production on Agar Plates with Tween-80

The culture medium containing (g/L): peptone 10, NaCl 5, CaCl_2_·2H_2_O (0.1), and agar (20) in distilled water was sterilized by autoclaving for 15 min (15 lb, 121 °C). After cooling the medium to 45 °C, 10 mL of Tween-80 was added as the final step. All microbial agar plates and the bacteria on tween-agar plates were incubated at 37 °C for 48 h. The enzyme activity was then determined based on the visual precipitation of calcium salts caused by the fatty acids released from the hydrolysis reaction [20].

### 2.3. Screening of Microbial Lipase Production on Agar Plates with Olive Oil

Phenol Red agar (pH 7.4) containing olive oil as a substrate for lipase-producing bacteria was prepared by dissolving (g/L) peptone 5, yeast extract 3, CaCl_2_ 1, and agar 15 in distilled water. The medium was adjusted to a pH of 7.4 using 0.1 M NaOH. After autoclaving for 15 min (15 lb, 121 °C) and cooling to 60 °C, 10 mL of olive oil and 10 mL of phenol red (1 mg/mL) were added. Bacteria on phenol red plates were incubated at 37 °C for 48 h. Because of the presence of fatty acids, a slight decrease in pH changes the phenol red color from pink to yellow, which is an indicator of the bacterial lipase activity [21].

### 2.4. Optimal Conditions for Lipase Production 

#### 2.4.1. Determination of a Suitable pH and Temperature

To optimize the pH value and temperature, bacterial culture medium containing (g/L): peptone (5) and yeast extract (5) was dissolved in distilled water and autoclaved for 15 min (15 lb, 121 °C), and 10 mL of olive oil was added after cooling the medium to 60 °C.

For optimal pH screening, bacterial cultures were exposed to different pH values (3, 5, 7, 9, and 11) and incubated at 30 °C. For optimal temperature screening, samples were incubated at different temperatures ranging from 5 to 75 °C (5 °C intervals) at pH 7. Produced enzymes were withdrawn at 8 h intervals over a period of 72 h.

#### 2.4.2. Determination of Optimal Carbon and Nitrogen Sources

Lipase, considered as an inducible enzyme, is affected by various carbon sources [22]. In addition to lipidic carbon sources, such as oil, triacylglycerols, fatty acids, tweens, and glycerols, other carbon sources, such as polysaccharides and sugars, influence the efficiency of lipase production. In this study, Tween-80, olive oil, glucose, maltose, and lactose were selected as carbon sources for lipase production. Pure isolates of lipase-producing bacteria were cultured and adjusted to 10^8^ CFU/mL in a medium (pH 7) containing 1% of each type of carbon source, 5 g yeast extract, and 5 g NaCl. 

Different nitrogen sources (1%) such as yeast extract, casein, peptone, ammonium nitrate, and potassium nitrate were added to the medium (pH 7) containing 5 g lactose and 5 g NaCl. All bacterial cultures were incubated at 30 °C and 120 rpm for 72 h. 

#### 2.4.3. Effect of Different Ions on Lipase Activity

Metal ions such as Ca^2+^, Zn^2+^, Cu^2+^, Fe^2+^, Mg^2+^, K^+^, Co^2+^, Mn^2+^, and Sn^2+^ at a concentration of 1mM were added to the enzyme solution to determine their effect on lipase activity [23]. After 1 h of incubation at 30 °C, enzyme solutions containing metal ions were subjected to a colorimetric assay to estimate the remaining activity.

#### 2.4.4. Effect of Agitation on Lipase Production

Bacterial cultures were established to study the effect of agitation: samples were incubated in a shaking incubator at 120 rpm, 150 rpm, and 160 rpm, and the other was maintained under static conditions at the same temperature of 30 °C.

### 2.5. Lipase Activity Assay

After optimizing the pH, temperature, carbon and nitrogen sources, bacterial strains were cultured in a fermentation medium that contained (g/L): yeast extract 5, NaCl 5, and glucose 5 at 120 rpm for 72 h. After 72 h, the bacterial cultures were centrifuged at 10,000 rpm for 10 min at 4 °C. Extracellular enzymes in the supernatant were used for enzymatic assays. 

Lipase activity was determined spectrophotometrically using p-nitrophenol palmitate (pNPP) as a substrate. The reaction mixture contained 700 μL pNPP solution and 300 μL lipase enzyme solution. The pNPP solution was prepared by mixing solution A (0.001 g pNPP in 1 mL of isopropanol) and solution B (0.01 g gum arabic, 0.02 g sodium deoxycholate, 50 μL Triton X-100 and 9 mL of 50 mM Tris-HCl buffer, pH 8). After incubating at 30 °C for 2 min, the absorbance of the mixture was immediately measured at 410 nm. One unit of lipase activity was defined as the release of 1 µmol of free p-nitrophenol per minute under the test conditions [24]. The enzyme concentration was calculated as follows:(1)$$\mathrm{Activity}\;\left(\frac{U}{mL}\right)=\frac{\left(p-\mathrm{nitrophenol}\right)\mu \mathrm{mol}}{\mathrm{mL}}x\frac{1}{\mathrm{incubation}\;\mathrm{time}}x\;\mathrm{dilution}\;\mathrm{factor}$$

### 2.6. Molecular and Phylogenetic Analysis

The 16S rRNA gene sequencing was performed on the genomic DNA extracted from the strong lipase-positive bacterial strains using the InstaGene Matrix kit (Bio-Rad, Seoul, Korea) according to the manufacturer’s instructions. As described by Frank et al. [25], amplification of the 16S rRNA gene was performed using PCR with primers 27F and 1492R. A multiscreen filter plate (Millipore Corp, Bedford, MA, USA) was used to purify the PCR product, which was then sequenced using the primers 518F (5-CCA GCA GCC GCG GTA ATA CG-3) and 800R (5-TAC CAG GGT ATC TAA TCC-3) with a PRISM BigDye Terminator v3.1 Cycle Sequencing Kit (Applied Biosystems, Foster City, CA, USA). This process was performed at 95 °C for 5 minutes. The product was cooled on ice for 5 min and analyzed using an ABI Prism 3730XL DNA analyzer (Applied Biosystems, Foster City, CA, USA). Finally, the nearly full-length 16S rRNA sequence was assembled using SeqMan software (DNASTAR Inc., Madison, WI, USA) [25]. The sequence similarity of each lipase-producing bacterial strain was determined by comparison with the available sequences in the GenBank database using the EZBioCloud server (http://ezbiocloud.net/ (accessed on 13 April 2021).) [26].

## 3. Results

### 3.1. Phylogenetic Analysis

Among the 55 bacterial strains exhibiting tolerance to contaminated environments, 17 strains were strong lipase-producers. In comparison to the traditional medium, the modified medium was considerably efficient in isolating functional bacterial strains under harsh conditions. Phylogenetically, potential lipase-producing strains belonged to 16 genera: *Bacillus, Lysinibacillus, Paenibacillus, Brevibacillus, Solibacillus Stenotrophomonas, Enterobacter, Klebsiella, Acinetobacter, Sporosarcina, Viridibacillus, Burkholderia, Exiguobacterium, Vagococcus, Bhargavaea*, and *Leclercia* compared to that of the popular *Bacillus* and *Paenibacillus* genera investigated from the traditional medium (Figure 1).

### 3.2. Quantitative Screening of Isolated Lipase-Producing Bacteria

Tween 80-based indicator is convenient and can easily investigate potential lipase producing bacteria based on the size of the zone around the colony [27]. The highest lipolytic activity was detected with a zone diameter of 22 mm by *Lysinibacillus* PL33, followed by *Lysinibacillus* PL35 (20 mm), *Paenibacillus* PL2 (19 mm), *Bacillus* L8 (18 mm), and 17 mm for *Bacillus* FW2, whereas *Stenotrophomonas* N-PL7 displayed the lowest lipolytic activity (10 mm) (Table 1).

### 3.3. Optimal Fermentation Conditions Affecting Lipase Production

#### 3.3.1. The Effect of Temperature on Lipase Production

In this study, the strongest active isolates, the highest lipase enzyme was produced by *Lysinibacillus* PL33, reaching 480 U/mL, followed by C with 420 U/mL, and *Bacillus* L8 with 390 U/mL at 35 °C. The strain *Paenibacillus* PL2 secreted a maximum of 400 U/mL, followed by strain *Bacillus* FW2, which produced 380 U/mL at 40 °C. Optimal lipase production by strains *Paenibacillus* L2 and *Stenotrophomonas* N-PL7 recorded at 30 °C reached 290 U/mL and 170 U/mL, respectively. Most of these bacterial strains grew and survived at 10 °C, except for strains *Enterobacter* PL4, *Paenibacillus* L2, and *Enterobacter* L7. *Lysinibacillus* PL33 and *Bacillus* FW2 were the strongest bacterial strains with lipase activity at 10 °C compared to the other strains and produced 120 U/mL. *Enterobacter* L7 and *Stenotrophomonas* N-PL7 did not grow at 40 °C and 45 °C, respectively. Other bacterial strains produced inadequate enzymes at 45 °C (Figure 2).

#### 3.3.2. The Effect of pH on Lipase Production

Most bacterial strains produced lipase at pH 5, except for *Paenibacillus* PL2 and *Stenotrophomonas* N-PL7, which could not survive at pH 5. Lipase yield increased from pH 6, was optimum at pH 7, and gradually decreased at pH 8. Only the strain *Stenotrophomonas* N-PL7 had no enzyme activity at pH 10, while others were capable of generating lipase, observed the highest amount of 200 U/mL produced by strain *Lysinibacillus* PL33. A similar pattern was observed in *Bacillus* FW2 and *Lysinibacillus* PL35, with an estimated 170 U/mL and 110 U/mL of *Bacillus* L8 (Figure 3).

#### 3.3.3. The Effect of Culture Variable Carbon and Nitrogen Sources

Among various carbon sources, Tween-80 was found to be the optimal carbon source for lipase activity by strain *Bacillus* FW2, accounting for a maximum of 350 U/mL, followed by 230 U/mL with glucose, while it decreased gradually with olive oil and maltose, with lactose being the most inefficient carbon source. Similar to the strain *Bacillus* FW2, the strains *Enterobacter* L7 and *Paenibacillus* L2 secreted the highest lipase peaking at 310 U/mL and 240 U/mL, respectively, in the presence of Tween-80, followed by olive oil, maltose, glucose, and lactose at 260, 150, 120, and 90 U/mL, respectively, for strain *Enterobacter* L7. Furthermore, for strain *Paenibacillus* L2 olive oil and maltose produced 150 U/mL and 120 U/mL of lipase, respectively, while glucose and lactose had the same effect on lipase production with only 70 U/mL. The remaining bacterial strains *Lysinibacillus* PL33, *Lysinibacillus* PL35, *Bacillus* L8, *Paenibacillus* PL2, and *Stenotrophomonas* N-PL7 obtained the maximum lipase yields in the presence of olive oil at 420, 400, 370, 350, and 150 U/mL, respectively. Maltose was the most suitable for lipase activity for *Lysinibacillus* PL33 than the other strains, accounting for 230 U/mL. In contrast, the presence of glucose decreased lipase production in *Stenotrophomonas* N-PL7 and *Enterobacter* L7 observed at 90 and 70 U/mL, respectively, compared to other carbon sources. Lactose was unsuitable for the strains *Stenotrophomonas* N-PL7, *Bacillus* FW2, *Paenibacillus* L2, and *Enterobacter* L7, which obtained only 50, 60, and 70 U/mL, respectively (Figure 4a).

Yeast extract was considered as the best nitrogen source for lipase activity by *Lysinibacillus* PL33 with the highest enzyme amount of 400 U/mL, followed by *Lysinibacillus* PL35, *Bacillus* L8, *Bacillus* FW2, *Bacillus* N-PL4, and *Paenibacillus* L2 at 380, 350, 290, 260, and 220 U/mL, respectively. In the presence of peptone, lipase production was optimal in *Enterobacter* L7 and *Paenibacillus* PL2, obtaining the same amount of lipase at 290 U/mL, *Enterobacter* PL4 at 260 U/mL, and *Stenotrophomonas* N-PL7 at 160 U/mL. Compared to the other strains, casein showed the highest efficiency in *Lysinibacillus* PL35 (270 U/mL), followed by *Bacillus* L8 (240 U/mL), and the same amount of 190 U/mL for both strains *Bacillus* FW2 and *Enterobacter* L7. However, the supplementation of potassium nitrate and ammonium nitrate in the culture medium did not significantly affect lipase activity compared to the other nitrogen sources (Figure 4b).

#### 3.3.4. The Effect of Metal Ions on Lipase Activity

Table 2 showed the effect of divalent ions on the activity of lipase of ten bacterial strains. The presence of Ca^2+^, Mg^2+^, and K^+^ in the culture medium enhanced the (practically doubled) lipase activity compared to that of the control in strains *Bacillus* FW2, *Lysinibacillus* PL35, *Lysinibacillus* PL33, and *Enterobacter* L7. Mg^2+^ stimulated the lipase effect on *Bacillus* L8, *Paenibacillus* L2, and *Bacillus* N-PL4. Mn^2+^ and Fe^2+^ improved lipase generation of *Lysinibacillus* PL33, *Lysinibacillus* PL35, *Bacillus* FW2, *Bacillus* L8, and slightly accelerated the enzymatic activity of *Bacillus* N-PL4 and *Enterobacter* L7. The results illustrated that Cu^2+^, Zn^2+^, Co^2+^, and Sn^2+^ inhibited the enzyme activity of all bacterial strains in this study, which demonstrated competitive inhibition with the enzyme, thereby reducing their catalytic activity.

#### 3.3.5. The effect of Agitation on lipase production

Lipase production, under shaking conditions in all bacterial strains, observed a 1.3–1.7 folds increase compared to static conditions by improving the rate of oxygen transfer, leading to easy dispersal of oil micelles into the microbial cell. The result of this study showed the similar trend to the previous studies (Figure 5).

## 4. Discussion

The published closest bacterial strains of ten strains in this study were investigated as non-lipase-producing bacteria. Moreover, there were limited studies working on the ability of lipase production of the members from the genus *Lysinibacillus* and *Paenibacillus* so far [28,29,30,31,32] (Table 3).

In general, the temperature for optimal lipase production rate is approximately 30–35 °C. However, other studies have demonstrated that bacteria may secrete the highest lipase enzyme at either low or high temperatures, such as at 10 °C or 40 °C [33,36]. One of the important targets of the research is to reduce the energy barrier of heating during the treatment process of wastewater, especially in the winter. Therefore, the results in this study demonstrated that temperatures ranging from 30 °C to 40 °C were the optimal conditions of lipase production for all bacterial strains. Particularly, *Lysinibacillus* PL33 and *Bacillus* FW2 were investigated as strong candidates that were capable of producing lipase at 10 °C that reached the goal of the research. In other studies, temperatures of 50 °C and 55 °C were considered optimum for effective lipase activity [37,38,39]. However, some *Bacillus* sp. produced the highest lipase at 60–80 °C [40,41,42,43]. While lipase from *Janibacter* sp. R02 has an optimal point at 90 °C [44].

*Bacillus* species in this study were able to produce lipase at pH 5. Another *Bacillus* sp. that exhibited maximum lipase activity at pH 5.6 was also investigated in another study [45]. In particular, in previous studies, bacterial strains from *Pseudomonas* and *Enterococcus* sp. produced lipase at acidic conditions of pH 3.5 and 4.6, respectively [46,47]. In this study, pH 7 was the optimal pH for lipase secretion of all bacterial strains. However, several studies investigated that a pH of 8–9 were the optimal pH values for lipase yield by *Bacillus* species [48,49,50,51].

It was demonstrated that carbon and nitrogen sources have a significant effect on lipase-producing efficiency. The remarkable point in most studies was the fact that the presence of lipidic carbon sources such as oils and fatty acids enhanced the lipase activities [52]. Such Tween 80 and olive oil were the optimal carbon sources that facilitate the activity of lipase of many bacterial species similar to the result in this study [53,54]. Organic nitrogen sources showed as great influence on the growth and promotion of bacterial lipase expression in both this study and the previous studies [55,56].

Metal ions play a critical role in maintaining the enzyme structure and activity [57]. As explained by Sharma et al., the formation of insoluble Ca salts of fatty acids released during hydrolysis may prevent the inhibition of enzyme activity [58]. Mg^2+^ and Ca^2+^ have been explored as the positive effect on lipase activity. It was explained that their contribution as the requirement of metalloprotein and the increase of thermal stability with the supplement of Ca^2+^ due to the presence of more binding sites [59]. However, Mg^2+^ partially reduced lipase activity in other studies [60,61,62]. Nevertheless, in a previous study, K^+^ had no significant effect on the enzyme activity of *Bacillus* sp. [63]. It was reported that Mn^2+^ can stimulate lipase activity [64] but in contrast, Mn^2+^ was an inhibitor of lipase in *Bacillus* sp. [48,65]. The decrease in lipase production was also observed in other studies caused by Fe^2+,^ Co^2+^, Zn^2+^ inhibition [59,66].

An agitation speed of 150 rpm was found to be optimal for lipase activity due to enhanced oxygenation in the culture medium [67]. However, increasing the agitation beyond 150 rpm may cause enzyme denaturation and damage the by-products due to the accumulation of hydrogen peroxide [68,69].

## 5. Conclusions

Conclusively, a modified culture method was introduced in this study investigating functional bacterial candidates that can tolerate various contaminants. By concocting an artificial mixture of metal ions and plastic-supplemented nutrient molecules from soil extract under shaking conditions, a number of lipolytic bacteria were explored, and their adaptability under harsh conditions was improved. A high lipase yield from these candidates may be of a potential value in other applications. Particularly, in the treatment of oily wastewater and oil-contaminated soil, the use of lipolytic bacteria may represent a green and economic alternative approach in the future. Therefore, further experiments in certain applications need to be carried out in lab-scale and large-scale as the next steps after the screening experiments in this study.

## Figures and Tables

**Figure 1 microorganisms-09-02590-f001:**
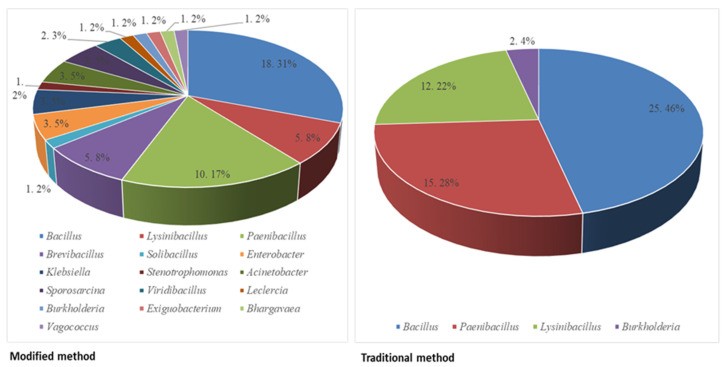
The diversity of bacterial strains investigated from the modified culture method compared to the traditional method.

**Figure 2 microorganisms-09-02590-f002:**
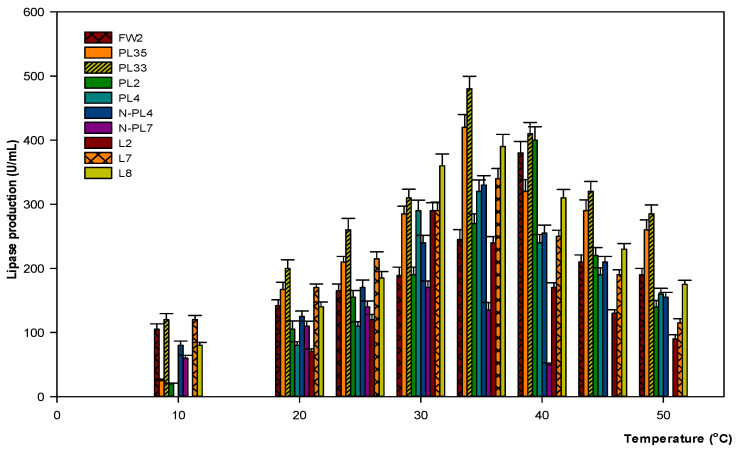
The effect of different temperatures on lipase production. Ten bacterial cultures were adjusted to a pH of 7 and incubated for 72 h. Experiments were performed in triplicate.

**Figure 3 microorganisms-09-02590-f003:**
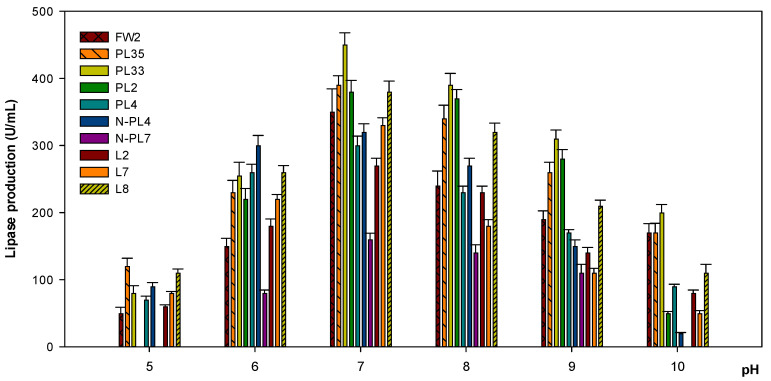
The effect of different pH values on lipase production. Bacterial cultures were incubated at 30 °C for 72 h. Experiments were performed in triplicate.

**Figure 4 microorganisms-09-02590-f004:**
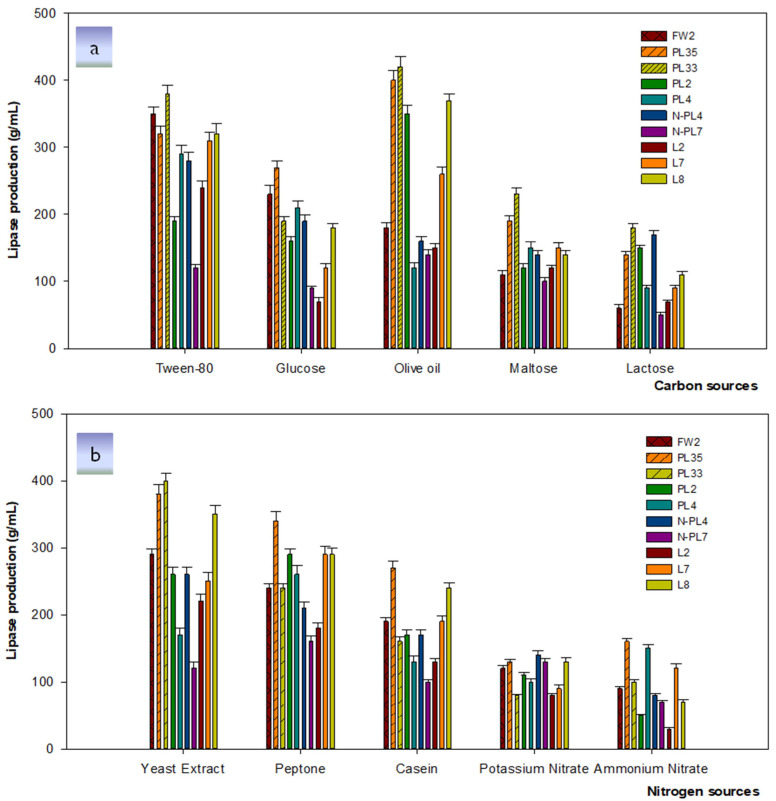
The influence of carbon (**a**) and nitrogen sources (**b**) on lipase activities. Samples were incubated 30 °C for 72 h. Experiments were performed in triplicate.

**Figure 5 microorganisms-09-02590-f005:**
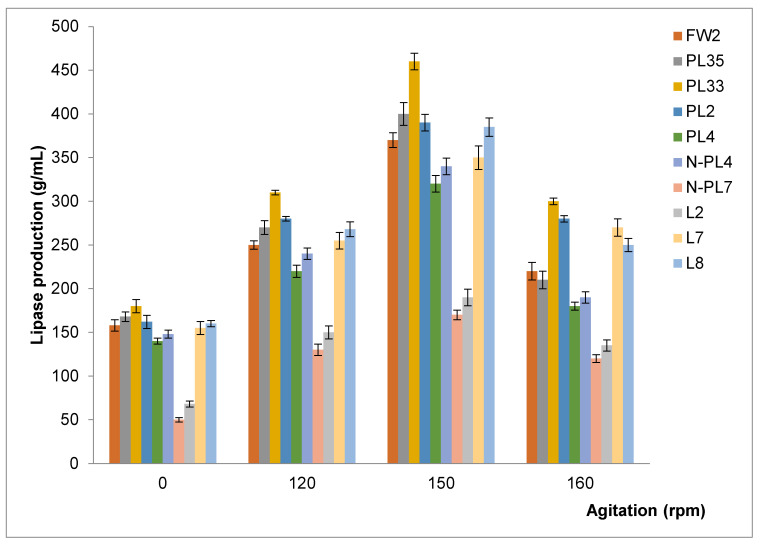
The effect of agitation on lipase production and growth of ten bacterial strains. Cultures were adjusted to a pH of 7, incubated at 30 °C at different agitations for 72 h. Experiments were performed in triplicate.

**Table 1 microorganisms-09-02590-t001:** Screening of lipolytic activity and lipase production of bacteria isolated from soil using modified culture method.

Number of Bacterial Strains	Isolates	Lipolytic Clearing Zone (mm)
1	*Lysinibacillus* PL33	22
2	*Lysinibacillus* PL35	20
3	*Paenibacillus* PL2	19
4	*Bacillus* L8	18
5	*Bacillus* FW2	17
6	*Enterobacter* L7	15
7	*Bacillus* N-PL4	14
8	*Enterobacter* PL4	13
9	*Paenibacillus* L2	12
10	*Stenotrophomonas* N-PL7	10

**Table 2 microorganisms-09-02590-t002:** Effect of metal ions on lipase activity of bacterial strains examined at 30 °C for 1 h.

No.	Bacterial Strains	Relative Activity Affected by Metal Ions (%)								
		Ca^2+^	Mg^2+^	K^+^	Fe^2+^	Zn^2+^	Cu^2+^	Co^2+^	Mn^2+^	Sn^2+^
1	*Lysinibacillus* PL33	100	100	98.2	100	55.7	65.2	35.6	100	65.8
2	*Lysinibacillus* PL35	99.2	97.7	96.8	98.7	45.3	59.5	41.3	99.5	68.9
3	*Paenibacillus* PL2	80.9	86.6	89.3	99.0	33.7	44.7	30.7	98.7	55.5
4	*Bacillus* L8	85.3	98.4	85.7	98.6	46.8	35.8	45.8	99.2	66.3
5	*Bacillus* FW2	98.8	100	100	100	58.3	49.3	55.6	99.6	68.7
6	*Enterobacter* L7	98.5	97.5	95.8	82.5	23.8	27.5	16.7	63.4	42.6
7	*Bacillus* N-PL4	90.3	100	85.3	80.9	34.9	55.8	22.5	79.8	43.9
8	*Enterobacter* PL4	83.4	86.2	81.9	60.5	14.7	26.4	31.3	56.7	31.2
9	*Paenibacillus* L2	84.6	99.4	78.7	75.8	22.5	13.9	23.8	60.3	20.8
10	*Stenotrophomonas* N-PL7	85.9	87.7	83.5	80.6	13.8	20.5	17.8	64.6	17.7

**Table 3 microorganisms-09-02590-t003:** The list of lipase-producing bacteria which belong to the same genus from the previous studies compared to ten bacterial strains in this study.

Microbial Source	Maximum Lipase Production (U/mL)	Application	References
*Lysinibacillus macroides* FS1	14.1	Biocatalyst for production of biodiesel	[28]
*Enterobacter cloacae Strain* POPE6	2.031	Apply to oil waste management	[29,30]
*Stenotrophomonas maltophilia*	4559	Degrade a harmful mycotoxin	[31,32]
*Bacillus cereus* HSS	285	Applied in waste water treatment	[33]
*Bacillus aryabhattai* SE3-PB	265.82	Potential biotechnological applications	[34]
*Bacillus licheniformis*	37.6	Industrial and environmental applications	[35]
*Lysinibacillus* PL33	480	Industrial, environmental, and biological applications	This study
*Lysinibacillus* PL35	420	Industrial, environmental, and biological applications	This study
*Paenibacillus* PL2	400	Industrial, environmental, and biological applications	This study
*Bacillus* L8	390	Industrial, environmental, and biological applications	This study
*Bacillus* FW2	380	Industrial, environmental, and biological applications	This study
*Enterobacter* L7	340	Industrial, environmental, and biological applications	This study
*Bacillus* N-PL4	330	Industrial, environmental, and biological applications	This study
*Enterobacter* PL4	320	Industrial, environmental, and biological applications	This study
*Paenibacillus* L2	290	Industrial, environmental, and biological applications	This study
*Stenotrophomonas* N-PL7	170	Industrial, environmental, and biological applications	This study

## Data Availability

The data used to support the findings of this study are available from the corresponding author upon request.

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
