# Peer review of "Investigation of Lipolytic-Secreting Bacteria from an Artificially Polluted Soil Using a Modified Culture Method and Optimization of Their Lipase Production"

_microorganisms, 2021, doi:10.3390/microorganisms9122590_

Round 1

Reviewer 1 Report

An interesting paper.  Just some corrections are needed (see in attachement).

Author Response

Dear Reviewer,

We would like to thank you very much for your valuable time and helpful comments on my manuscript to make it better before acceptance. We have revised point by point in the manuscript and responded to your comments. Please find our responses in the attached file.

All the bests,

Woojin Chung

Reviewer 2 Report

Dear Authors,

the manuscript is particularly interesting because the use of lipases is now a biotechnological reality. However, regarding water treatment (e.g. waste water from olive mills), the authors must clarify the following:
It has been reported by the authors that temperatures between +30 ° C and +40 ° C are the optimal conditions for lipase production for all bacterial strains, in particular, Lysinibacillus PL33 and Bacillus FW2 have been indicated as strong candidates in capable of producing lipase at 10 ° C, in other studies, temperatures of +50 ° C and +55 ° C were considered optimal for effective lipase activity. However, some Bacillus sp. produced the highest lipase 316 at +60 or +80 ° C, while Janibacter sp. R02 has an optimal temperature of +90 ° C, given the high temperatures necessary for the production of high quantities of lipase, the authors have not clarified how it is possible to produce high quantities of lipase at interesting costs for industrial applications or for the treatment of waste water from the mills given the high volumes treated. Ultimately, the authors must make the reader understand if experiments have been carried out in the field, or experiments applied to industrial realities and for the detoxification of waste water or if this manuscript deals with studies without real applications in the field or basic studies that require subsequent "scaling up" and related application considerations.

Best Regards

Author Response

(The authors gave the same response as above.)
